# Histovariability and Palaeobiological Implications of the Bone Histology of the Dromornithid, *Genyornis newtoni*

**Anusuya Chinsamy** [1,*] and **Trevor H. Worthy** [2]

1   Department of Biological Sciences, University of Cape Town, Private Bag X3, Rhodes Gift,
    Cape Town 7700, South Africa
2   College of Science and Engineering, Flinders University, GPO 2100, Adelaide 5001, Australia;
    trevor.worthy@flinders.edu.au
*   Correspondence: anusuya.chinsamy-turan@uct.ac.za; Tel.: +27-21-650-3604

**Abstract:** The bone microstructure of extinct animals provides a host of information about their biology. Although the giant flightless dromornithid, *Genyornis newtoni,* is reasonably well known from the Pleistocene of Australia (until its extinction about 50–40 Ka), aside from various aspects of its skeletal anatomy and taxonomy, not much is known about its biology. The current study investigated the histology of fifteen long bones of *Genyornis* (tibiotarsi, tarsometatarsi and femora) to deduce information about its growth dynamics and life history. Thin sections of the bones were prepared using standard methods, and the histology of the bones was studied under normal and polarised light microscopy. Our histological analyses showed that *Genyornis* took more than a single year to reach sexual maturity, and that it continued to deposit bone within the OCL for several years thereafter until skeletal maturity was attained. Thus, sexual maturity and skeletal maturity were asynchronous, with the former preceding the latter. Our results further indicated that *Genyornis* responded to prevailing environmental conditions, which suggests that it retained a plesiomorphic, flexible growth strategy. Additionally, our analyses of the three long bones showed that the tibiotarsus preserved the best record of growth for *Genyornis*.

**Keywords:** Australia; Pleistocene fossil bird; dromornithid; *Genyornis*; bone histology; osteohistology





## 1. Introduction

It is well recognised that, during life, the bones of vertebrates are living tissues that record various aspects of their life history and biology, e.g., [1–3]. A host of studies have shown that even after millions of years of fossilisation, the bone microstructure is often well preserved, e.g., [1,2,4–6]. Thus, by studying the bone histology of extinct vertebrates, various inferences into their biology can be made.

*Genyornis newtoni* was a giant flightless galloansere bird that belonged to the Dromornithidae [7,8]. The dromornithids first appeared in the fossil record in the Eocene and reached the height of their diversity in the middle Miocene. By the late Oligocene, the dromornithids had already attained a large size and were flightless with reduced wings, a morphology they retained for the next 25 ma [7]. However, by the Pleistocene, *Genyornis newtoni* [9,10] was the only surviving member of the family.

*Genyornis newtoni* was widespread, though usually rare, in south eastern Australia in the mid-to-late Pleistocene and remains the only species of dromornithid for which individual skeletal assemblages are known. *Genyornis* went extinct in the late Pleistocene along with many other megafaunal animals about 50–40 Ka [7,11]. Aside from some general reconstructions of *Genyornis* as a medium-sized dromornithid (180–250 kg) that stood about 2–2.5 m tall, little is known about its biology. However, the well-defined bimodality in skeletal measurements is attributed to marked sexual dimorphism [12], where the males are assumed to be the larger sex, as was demonstrated for its larger relative *Dromornis stirtoni* for which medullary tissue was found in examples of the smaller morph [13]. Here, we

investigated its bone histology to deduce various aspects of its biology, particularly to infer how their growth dynamics allowed them to reach their large body size.

Modern birds generally, grow to adult body size within a single year [14,15]. Even large modern birds such as *Sagittarius serpentarius*, the secretary bird [16,17], and the largest modern bird, *Struthio camelus*, the ostrich [16,17], reach adult body size within a single year. Thus, the bones of modern birds generally have uninterrupted growth until an adult body size is reached, which is usually coincident with the attainment of sexual maturity [4]. However, once their growth rate slows down (usually upon or close to sexual maturity), their rate of bone deposition (osteogenesis) slows down, and a different type of bone tissue develops in their compacta. This outer band of tissue, called the outer circumferential layer (OCL) [18], often shows lines of arrested growth (LAGs) therein, indicating that these birds experience periodic arrests in growth as they slowly accrete bone for a few more years until skeletal maturity is reached. Although most modern birds grow like this, there are exceptions to such rapid growth rates among birds. This is particularly the case among insular birds such as the Apterygiformes (kiwi) [19], the Dinornithiformes (moa) [20] and the Aepyornithidae (elephant birds) from Madagascar [5]. Among the aepyornithids, *Vorombe titan* is largest, and like the large moa (Emeidae and Dinornithidae) [20], it takes several years to reach adult body size. It has been suggested that these birds grow much slower than other birds, because they are island birds without the pressure of mammalian predators [15]. Long-lived birds possibly also experience slower growth rates, but this needs to be verified—a single bone of a parrot (*Amazona amazonica*) had a growth mark therein [21], but there are no details of the approximate age of the individual.

The aim of our study was to assess the histology of *Genyornis* to decipher information pertaining to their growth dynamics and life history. As different bones would be studied, we also assessed the histovariability of the long bones studied, and we ascertained which elements were more reliable for growth assessment.

## 2. Materials

The *Genyornis* bones we studied were recovered from late Pleistocene lacustrine deposits at Lake Callabonna (48–45,000 years ago) and from mid-to-late Pleistocene fluvial deposits along Cooper Creek and Billeroo Creek [22] Figure 1. These deposits are assumed to have sampled palaeoenvironments of arid grassland/shrubland with some trees along watercourses, although current palaeoenvironmental studies are underway and should shed more light on the palaeoecology of these sites.

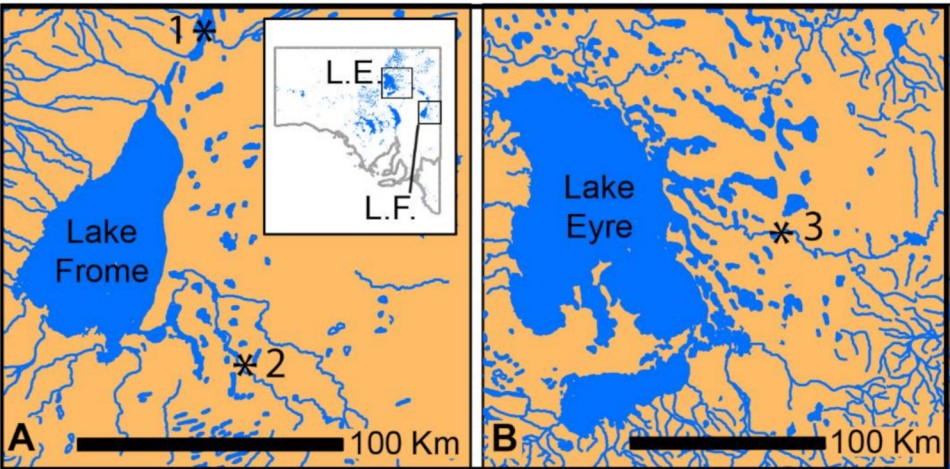

**Figure 1.** Sites in South Australia (inset) where the sampled bones of *Genyornis* were collected. (**A**) Lake Frome area (L.F.) and (**B**) Lake Eyre area (L.E.). 1, Lake Callabonna; 2, Billeroo Creek sites; 3, Cooper Creek sites (Malkuni Waterhole and Site 73).



Fifteen specimens of *Genyornis* were studied: eight tibiotarsi, three tarsometatarsi and four femora. As far as possible, standard measurements were taken of the limb bones studied, and the least shaft circumference of the tibiotarsus was measured in all specimens where it was preserved. The latter ranged from 134 to 146 mm (Table 1). For specimen SAM P.53833, we were able to sample different bones of the skeleton: a femur and tibiotarsus. All the other successfully sampled bones were from different individuals. In some specimens, parts of the shaft were opportunistically sampled (through natural breakages, etc.), but the majority of the specimens were core-sampled in the midshaft region to obtain the best possible track record of growth (see details in Table 1) [1].

**Table 1.** Details of the *Genyornis* specimens sampled. All sites are in South Australia. Unsampled elements of individuals are listed for their ability to reveal the size of the individual. Shading is used to show associated bones of individuals.

| Catalogue No. | Locality | Field ID | Element | Histology Sample | Comment | Sex |
|---|---|---|---|---|---|---|
| SAM P.25017 | Cooper Creek | | Distal left tibiotarsus | y | distal width, 85 mm; minimum shaft circumference, 140 mm; sampled section of shaft 1/3 length from distal end. | Female? |
| SAM P.53826 | Callabonna | Geny 1A | Distal right tibiotarsus | y | distal width, 88 mm, shaft inflated by salt degradation; sampled caudal facies. | Female? |
| SAM P.53826 | Callabonna | Geny 1A | Right tarsometatarsus | | max proximal width, 105 mm; min shaft width, 45 mm; distal width, 99 mm | |
| SAM P.53831 | Callabonna | Geny 9A | Right tarsometatarsus | y | Length trochlea III-cotyla lateralis, 358 mm; min shaft diameter, 50 mm; distal width, 110 mm; sampled medioplantar facies. | Male |
| SAM P.53832 | Callabonna | Geny 9B | Left tarsometatarsus | y | min shaft diameter, 44 mm; distal width, 95 mm; sampled medioplantar facies. | Female |
| SAM P.53833 | Callabonna | Geny 10 | Right femur | y | midshaft diameter, 76 mm; max distal width, 154 mm, sampled mid-caudal facies | |
| SAM P.53833 | Callabonna | Geny 10 | Left tibiotarsus | y | distal width, 86 mm; total length, 610 mm; minimum shaft circumference, 144 mm; sampled mid-medial facies | Female? |
| SAM P.53833 | Callabonna | Geny 10 | Right tarsometatarsus | | length, 347 mm; TL2, 355 mm, distal width medial-lateral, 102; min shaft width, 39 mm; max distal width, 92 mm | |
| SAM P.54333 | Cooper Creek | Geny C | Distal left tibiotarsus | y | distal width, 91 mm; sampled anterolateral facies | Female? |
| SAM P.54334 | Cooper Creek | Cooper Creek 73-B | Distal left tibiotarsus | y | distal width, 92 mm; minimum shaft circumference, ±140 mm; sampled section midshaft | Female? |
| FU2750 | Callabonna | CB2018-98 | part left tarsometatarsus | y | min shaft diameter, 38 mm; sampled medial facies | Female? |
| FU2755 | Callabonna | CB2018-75 Ind 1 | Right femur | y | shaft width, 87 mm; small indiv; sampled mid-caudal facies | |
| FU2755 | Callabonna | CB2018-75 Ind 1 | Right tibiotarsus | y | distal width, 88 mm; minimum shaft circumference, 144 mm; small indiv; 2 samples distomedial facies. | Female? |
| FU2755 | Callabonna | CB2018-75 Ind 1 | Left tarsometatarsus | | distal width, 90 mm; min shaft width, 40 mm; max proximal width, 102 mm; small indiv. | |
| FU2756 | Callabonna | CB2018-75 Ind 2 | Crushed left and right femora | | | |
| FU2756 | Callabonna | CB2018-75 Ind 2 | Right tibiotarsus | y | distal width, 96 mm; minimum shaft circumference 146 mm; big indiv; sampled midshaft medial facies | boundary-male/female? |
| FU2756 | Callabonna | CB2018-75 Ind 2 | Left tarsometatarsus | | min shaft diameter, 46 mm; max proximal width, 109 mm; max distal width, ±105 mm | |
| FU2760 | Callabonna | CB2018-75 Ind 3 | Right femur | y | midshaft width, ~66 mm; sampled mid-caudal facies | Male |
| FU2758 | Billeroo Creek | NA | Left femur | y | midshaft width, 70 mm; surface texture is porous; crista trochanteris is not fully formed; sampled mid-caudal facies. | young indiv? |
| FU2759 | Billeroo Creek | NA | Distal left tibiotarsus | y | Sampled medial facies | Small-female? |

The sampled bones were embedded in resin and were thin-sectioned according to standard petrographic methods [23]. They were then sectioned along the midline, resulting in two blocks labelled as A and B (which permitted the investigation of the histology closest to the neutral region, i.e., the area least affected by remodelling changes [1]). Four thin sections were prepared from these blocks (AI, AII; BI, BII). The sections were studied under petrographic microscopes (Nikon Eclipse E200 with a Nikon DS-Fi1 camera or a Zeiss Ax10 Lab.A1 with an Axiocam camera). All thin section preparations and photomicroscopies were performed at the University of Cape Town, South Africa.

Given that the dromornithid *Dromornis stirtoni* has a marked bimodality with females shown to be the smaller morph through the presence of medullary bone [13], its size has the potential to establish the sex of *Genyornis* bones. *Genyornis newtoni* shows a nonoverlapping bimodal size distribution [12], where tibiotarsi with a least shaft circumference of 137–150 mm are presumed females and those with values greater than 150 mm are males (assuming that dimorphism is the same in species of *Dromornis* and *Genyornis*, which is reasonable as males are uniformly larger birds in all extant Galloanseres). However, we used the totality of measurements from associated bones of an individual to infer sex, as for several individuals, the minimum shaft circumference was not measurable, but values for the tibiotarsal distal width or width for the femora and tarsometatarsi of the same individual were available. These revealed two individuals that were much larger than those biggest based on tibiotarsal shaft circumference and, thus, were inferred as males (Table 1).

Here, we followed the traditional histological terminology sensu [1,18,24]. Although we used the orientation of the canals in the bones as a proxy for the orientation of the vascular canals, we are aware that this does not accurately reflect the orientation of the blood vessels therein [15].

## 3. Results

The histology of several examples of hind limb elements are described and summarized as follows: eight tibiotarsi, four femora and three tarsometatarsi.

### 3.1. Tibiotarsi

3.1.1. FU2759, Billeroo Creek, F

Thin sections of this specimen showed that the bone tissue had experienced some taphonomic damage by infiltration of the surrounding minerals and sediment into the bone, concordant with its fluvial deposition. Thus, most of the specimen showed more extensive damage to the peripheral parts of the compacta, whereas the internal bone tissues were better preserved. Figure 2 shows the alteration in the bone microstructure caused by this infiltration damage. Nevertheless, it is evident that the bone has vascular canals right up to the margin, and there appears to be no distinctive slowing down in the rate of bone deposition (Figure 2A). Deeper in the compacta, the tissue continues to be primary periosteal bone with abundant vascular canals in a woven bone matrix. The vascular canals tend to be a mixture of longitudinal and short circumferentially oriented canals in a laminar arrangement (Figure 2B). In the innermost region of the compacta, closest to the medullary cavity, the bone tissue has a different appearance; here, the bone is still fibrolamellar tissue, but the vascular canals have a much more disarrayed arrangement with a predominantly longitudinal-to-radial arrangement in the woven bone matrix (Figure 2C. Throughout the compacta, secondary osteons are rare, sparse erosion cavities occur and no growth marks (i.e., LAGs or annuli) are apparent in the compacta (Figure 2).

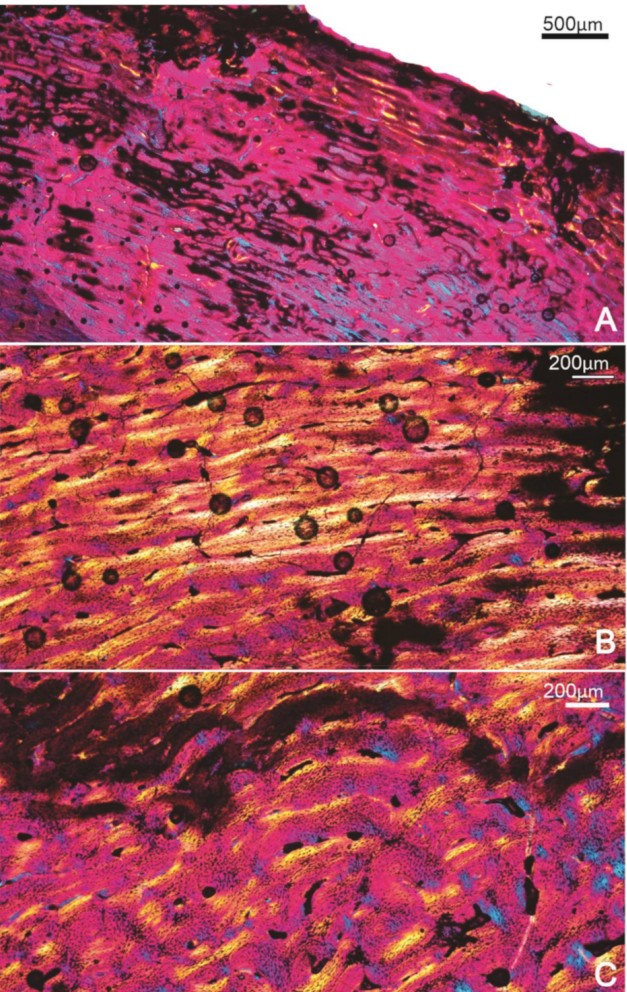

**Figure 2.** Specimen FU2759; tibiotarsus. (**A**) Outermost region of the compacta showing some diagenetic features but no change in the rate of bone formation, and vascular canals right up to the edge of the bone wall. (**B**) Mid-cortical region showing the circumferential organisation of the vascular canals. (**C**) Deep cortex of the compacta showing the more haphazard arrangement of the vascular canals within a woven bone matrix.

### 3.1.2. Specimen SAM P.54333, Cooper Creek (Geny C), ?F

The histology of this specimen is reasonably well preserved with the entire compacta from the periphery to the endosteal region preserved (Figure 3). Localised differences in the orientation of the vascular canals, and the nature of the bone tissue are evident; the outermost bone tissue, which is the most recently formed bone, comprises a narrow (~250 μm) band of avascular lamellar tissue, the OCL (Figure 3A,B). Preceding the OCL, there is a band of reticular bone tissue (Figure 3A,B), whereas deeper into the compacta, the tissue changes to a mix of plexiform to circumferential laminar bone tissue (Figure 3A,B). In the perimedullary region, several enlarged cavities are evident, and there are several secondary osteons that occur right up to the margin of the medullary cavity (Figure 3A), and in some areas, cancellous tissue extends into the medullary cavity. A narrow layer of lamellar bone, the inner circumferential layer (ICL), lines the medullary cavity in localised areas (Figure 3C). Except for this region, the rest of the cortex comprises mostly primary bone tissue, although there are a few scattered secondary osteons. In the thickest part of the bone wall, the innermost bone tissue comprises FBL with more longitudinal and circumferentially arranged primary osteons (Figure 3C). It is likely that this tissue is bone formed during the early stages of ontogeny.

### 3.1.3. Specimen SAM P.54334, Cooper Creek site 73B, F

The bone wall is incompletely preserved but there is enough of the compacta visible to make a histological description. The outermost part of the bone wall has a layer of lamellar bone, the OCL, that varies in thickness around the bone wall. Several lines of arrested growth (LAGs or rest lines) are visible in the OCL (Figure 4A). Below the OCL is a layer of richly vascularised bone tissue that varies from a reticular to a plexiform type of bone tissue with several growth marks in the form of annuli (Figure 4A,B). Note that the innermost and outermost annuli appear to be quite wide (Figure 4A). In the perimedullary region, there occurs a region of well-vascularised bone tissue that appears to be bone formed during the early stages of ontogeny (Figure 4A,C). Closer to the medullary cavity, there is a large amount of secondary reconstruction (Figure 4A) where numerous secondary osteons are visible, but these do not reach a high density. In the endosteal region, extensive remodelling of the compacta is evident, and there are large excavations into the compacta (Figure 4A). Small patches of ICL are visible in localised areas (Figure 4A).

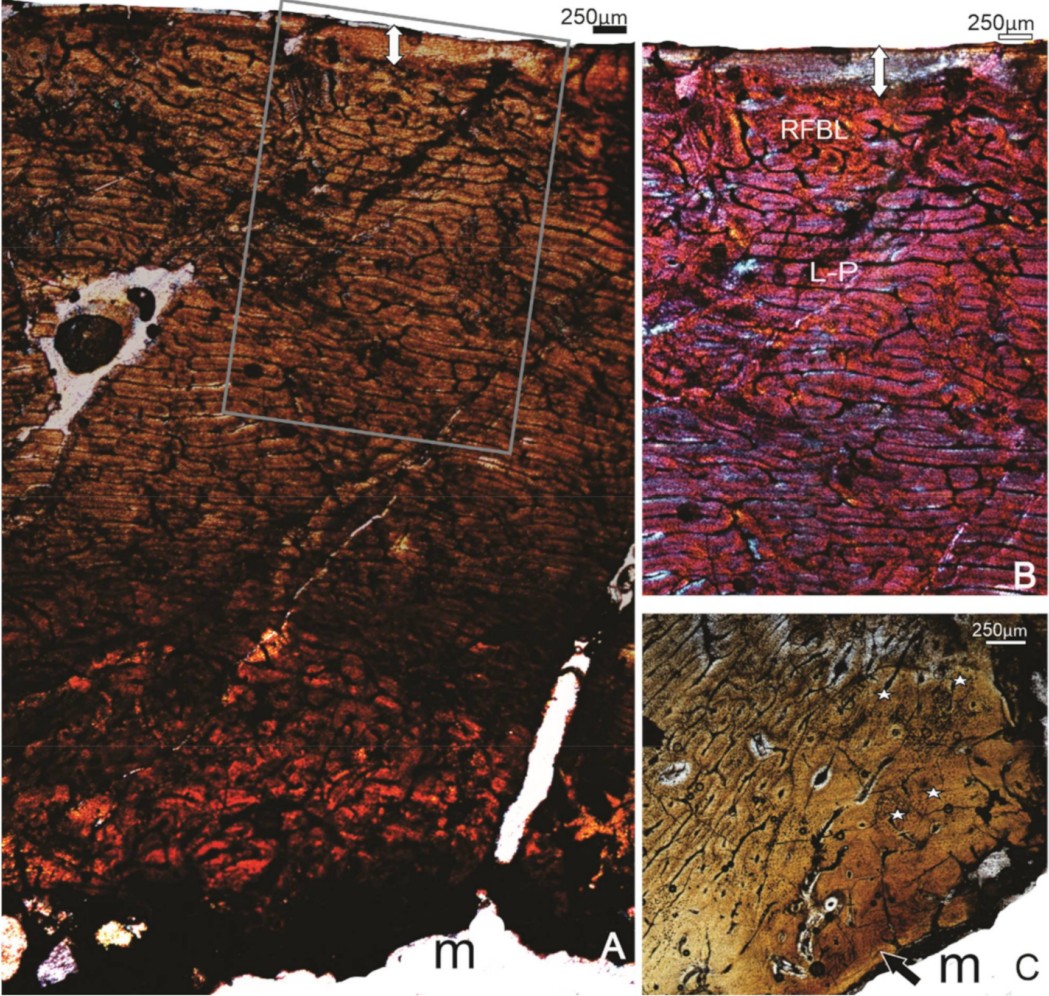

**Figure 3.** Specimen SAM P.54333; tibiotarsus. (**A**) Overview of the histology of the bone wall. (**B**) Higher magnification of the framed region in (**A**), showing the OCL (double-headed arrow) below which is a band of reticular organised FBL(RFLB), and the more laminar-plexiform (L-P) organised bone tissue deeper in the compacta. (**C**) A view of the perimedullary region showing a narrow ICL (arrow), and some remnants of the early formed reticular bone tissue, secondarily enlarged vascular canals and a few secondary osteons (stars). m, medullary cavity.

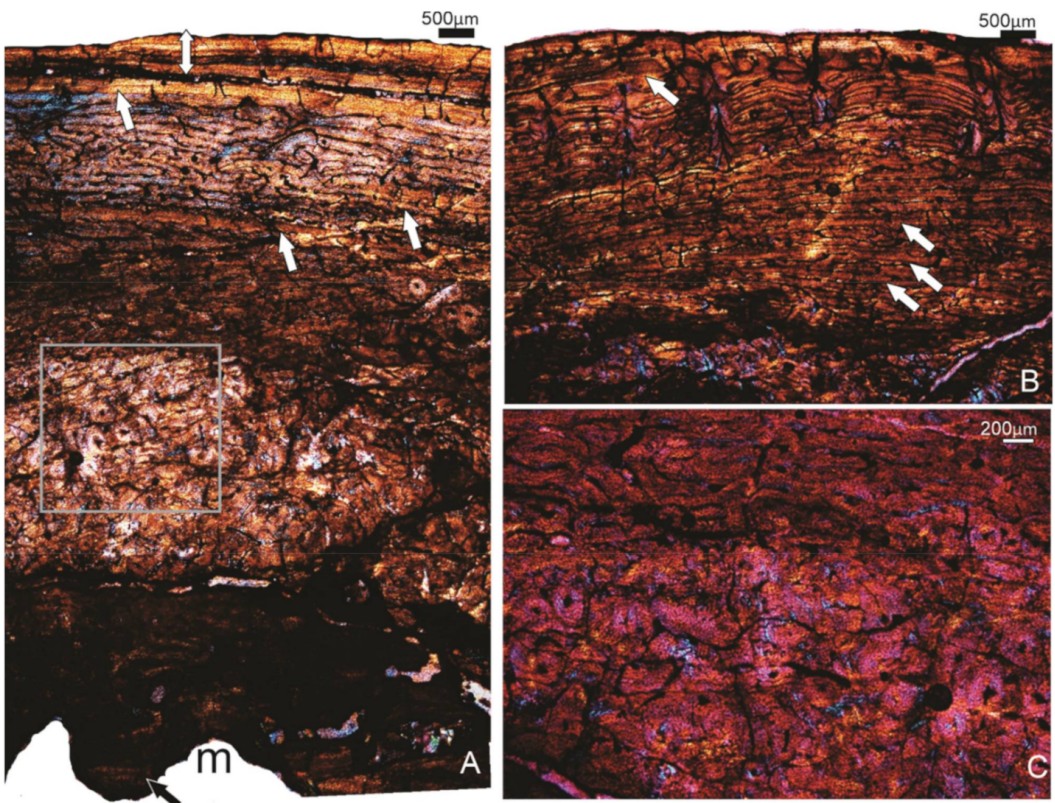

**Figure 4.** Specimen SAM P.54334; tibiotarsus. (**A**) Overview of the compacta showing an OCL in the peripheral region (double-headed arrow), several growth marks (annuli) (arrows) and a resorptive medullary (m) margin. The black arrow points to remnants of an ICL. (**B**) A different part of the compacta showing four growth marks (arrows) in the compacta. (**C**) Higher magnification of the framed region in (**A**) showing the reticular FBL bone tissue formed during early ontogenetic stages. m, medullary cavity.

### 3.1.4. Specimen SAM P.53826, Callabonna Geny 1A, F

Unfortunately, we were unable to retrieve a full core from this specimen, and only the outer part of the bone wall was preserved. Microscopical examination revealed that the bone is badly fractured, concordant with salt damage, but a distinct OCL is visible with at least three (perhaps four) growth marks in the form of narrow annuli present. Below the OCL, several circumferentially organised vascular canals are visible.

### 3.1.5. Specimen FU2756 Callabonna CB2018-75, Indiv 2, ?M/F

The outermost compacta consist of a wide layer of poorly vascularised lamellar bone tissue that forms the OCL, which is interrupted by at least 10 growth marks (LAGs) (Figure 5A). Below this outer band of tissue, the cortex consists of a more richly vascularised, more laminarly organised FLB tissue (Figure 5B). A narrow annulus with a LAG interrupts the deposition of this tissue (Figure 5B). Deeper in the compacta, the bone formed during the early stages of ontogeny is visible and appears to be FLB with mainly longitudinally and reticular organised vascular canals (Figure 5C). Some of these vascular canals have been enlarged by secondary reconstruction, and in some, there are secondary deposits of centripetally formed lamellar bone which form secondary osteons (Figure 5C).

### 3.1.6. Specimen FU2755, Callabonna CB2018-75, Indiv 1, F

The bone tissue of this tibiotarsus was not well preserved. The outer cortex was not sampled, so the most recently formed bone tissue cannot be described (and we cannot assess whether or not an OCL is present). The part of the compacta that was preserved

shows a richly vascularised primary compacta with predominantly longitudinal and reticular-oriented vascular canals.

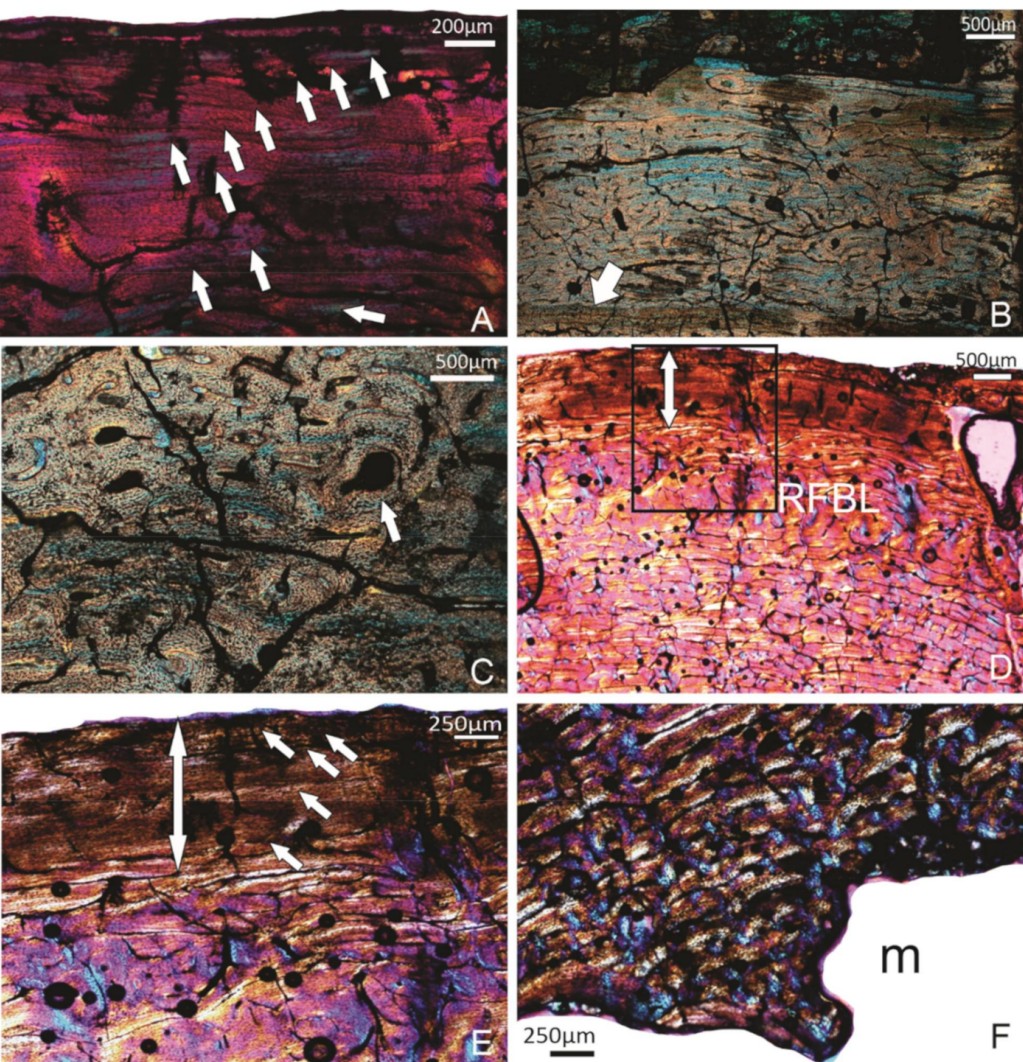

**Figure 5.** (**A–C**) Specimen FU2756; tibiotarsus. (**A**) The OCL with at least 10 LAGs (arrows). (**B**) A deeper part of the compacta showing the laminar organised FBL bone tissue, and a single annulus with a LAG (arrow). (**C**) FBL bone tissue with more longitudinal and reticular organised vascular canals. Arrows point to the lamellar bone tissue in the process of being deposited around an enlarged cavity. (**D–F**) Specimen SAM P.25017, tibiotarsus. (**D**) A general view of the compacta. The double-headed arrow indicates the OCL. RFBL indicates the band of reticular organised FBL bone tissue that precedes the OCL. (**E**) Higher magnification of the framed region of (D). The OCL with 5 LAGs (arrows). (**F**) The innermost part of the compacta being actively resorbed. m, medullary cavity.

### 3.1.7. Specimen SAM P.25017, Cooper Creek, Malkuni WH, F

The well-preserved compacta of this tibiotarsus shows that it is heavily vascularised with a distinctly wide OCL comprised of lamellar bone tissue with at least five LAGs (Figure 5D,E). A few blood vessels occur in the OCL, but overall, it is much more sparsely vascularised than the underlying bone tissue. Preceding the OCL is a region comprising reticular primary bone tissue, whereas deeper parts of the compacta have a more plexiform arrangement. Overall, the bone wall appears to be primary in nature, but there are some scattered secondary osteons visible. In parts of the compacta, at least three narrow lamellar deposits interrupt the rapid deposition of bone. The perimedullary region of the cortex is uneven due to extensive resorption which cuts into the original early formed primary compacta of the bone wall (Figure 5E).

### 3.1.8. Specimen SAM P.53833, Callabonna, Geny 10, F?

A striking feature of the compacta is the presence of two distinct bands of lamellar bone towards the outer cortex (Figure 6A,B). These wide bands of more slowly deposited tissue are separated by an almost equally thick region of fibrolamellar bone tissue. Note that there are several incursions of blood vessels through both the lamellar layers, but overall, these layers are not as well vascularised as the tissue below. The outer band of lamellar tissue has about three LAGs, and this appears to be the OCL (Figure 6A,B). Below the more inner wide band of lamellar tissue, a narrow annulus is visible, and perhaps another, but the latter cannot be followed around the compacta (Figure 6A). Deeper in the cortex (the inner 40% of bone wall) is extensively reconstructed and, in places, reaches dense Haversian bone levels where even interstitial bone is secondary (Figure 6A). Many erosion cavities are visible, and there are many examples of these connecting with one another to form even larger cavities (Figure 6C). No ICL is present.

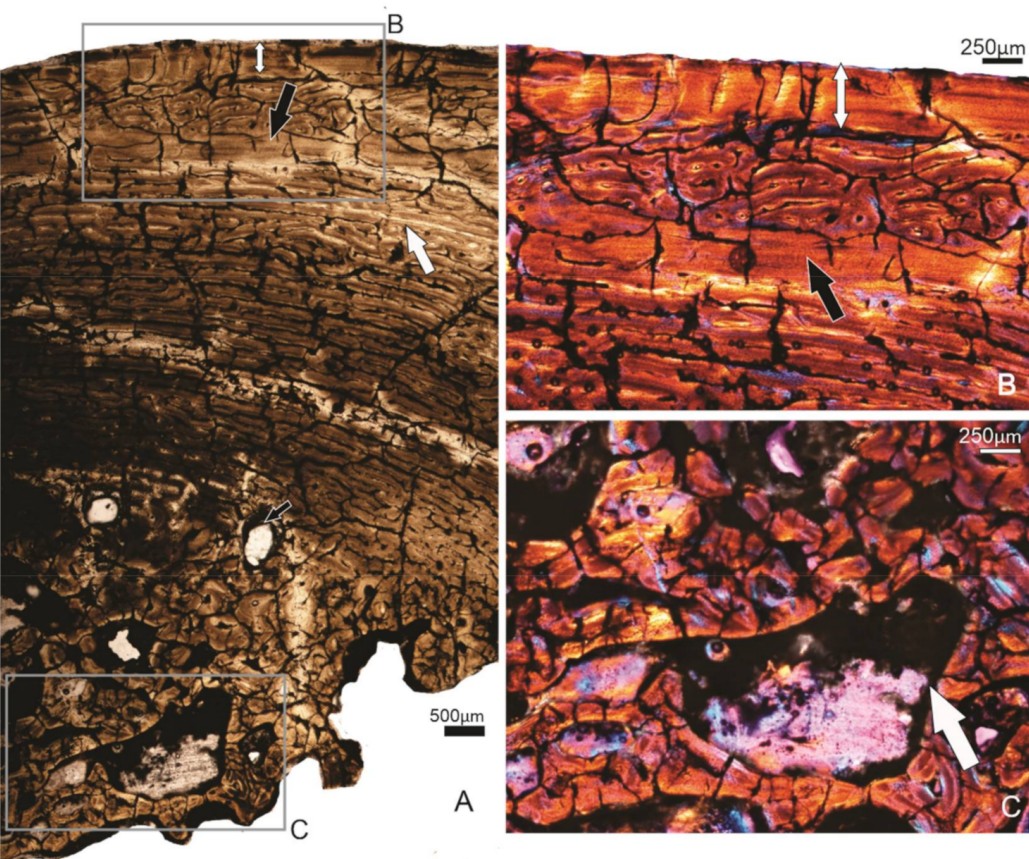

**Figure 6.** Specimen SAM P.53833, tibiotarsus. (**A**) Overview of the compacta. Subperiosteally, there is a wide OCL present (double-headed arrow), and further in the compacta, another wide band of lamellar tissue occurs (black arrow), below which is a narrow annulus (white arrow). Deep parts of the compacta show secondary remodelling, and several large erosion cavities are visible (small black arrow). (**B**) Higher magnification of the framed region of (A), showing the OCL and the wide inner band of lamellar tissue. (**C**) Higher magnification of the framed region in (A), showing the extensively secondary remodelled perimedullary region.

### 3.1.9. Summary of the Tibiotarsus Histology

The eight tibiotarsi studied here provide information about the overall growth of this skeletal element. There are obvious changes that are related to ontogenetic age, such as the development of the OCL, which clearly indicates that specimen FU2759 is the youngest of all the tibiotarsi, whereas specimen SAM P.54333 appears to be a slightly older individual that has already begun slowing down its overall rate of growth and only has a narrow OCL present. Besides the onset of OCL formation, the latter specimen also shows much more

secondary remodelling as compared to specimen FU2759, but less than the other tibiotarsi studied. This suggests that secondary remodelling increases with age. All the other specimens have a relatively wide OCL with varying numbers of LAGs present—specimen FU2756 has the widest OCL with at least 10 LAGs present (although it is uncertain whether any of these are double or triple LAGs). Specimen SAM P.54334 clearly shows at least four annuli that precede the deposition of the OCL, and interestingly, the deepest annulus and the last one appear quite wide.

### 3.2. Femora

### 3.2.1. Specimen FU2758 Left Femur, Billeroo Creek

Overall, the bone wall is not very well preserved, but histological details are discernible. The compacta is richly vascularised, and there is a thin band of lamellar bone tissue visible along the outermost peripheral part of the bone wall (Figure 7A). Below this, there are mainly longitudinally arranged primary osteons. In the mid-cortex, a few secondarily enlarged erosion cavities are evident, and a few completely formed secondary osteons can be seen (Figure 7B). A narrow ICL is present in places, suggesting that medullary expansion has been completed. No growth marks are visible anywhere in the compacta.

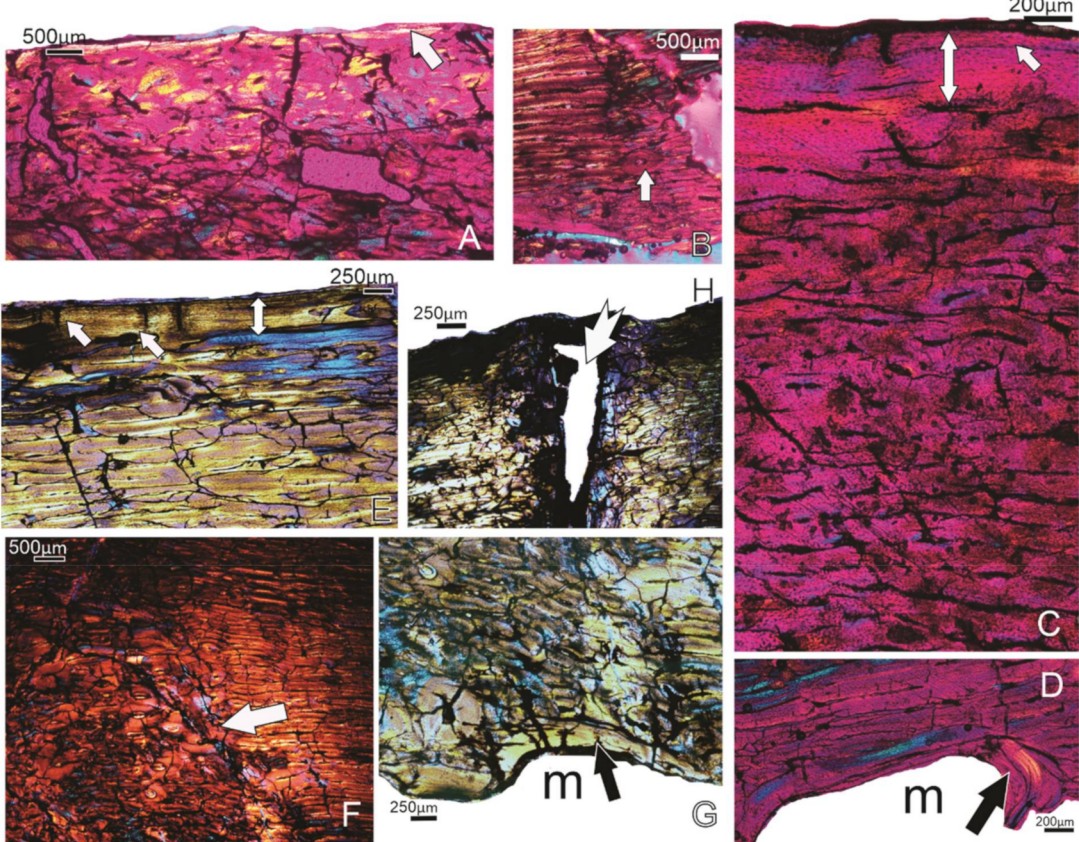

**Figure 7.** (**A**,**B**) Specimen FU2758; femur. (**A**) The white arrow points to the narrow deposit of lamellar bone tissue subperiosteally. (**B**) Some secondary osteons (arrow) are visible in the compacta. (**C**,**D**) Specimen FU2760; femur. (**C**) The well-vascularised compacta, and an OCL (double-headed arrow) with one LAG (arrow). (**D**) The deposition of an ICL (black arrow). (**E**–**H**) Specimen SAM P.53833; femur. (**E**) The OCL (double-headed arrow) and the laminar bone tissue that precedes it. (**F**) The tract of secondary osteons (arrow) that are coincident with the linea. (**G**) The uneven perimedullary margin, and the narrow ICL (arrow) in places. (**H**) The white arrow shows the entry of the nutrient foramen into the femur. Notice the changes in the orientation of the bone tissue along the margins of the canal. m, medullary cavity.

### 3.2.2. Specimen FU2760, Callabonna CB2018-75, Indiv 3, M

The compacta comprise predominantly short circumferentially organised vascular canals in a laminar arrangement (Figure 7C). An OCL is present, and a distinct LAG is visible therein (Figure 7C). In the perimedullary region, a well-developed ICL is visible, which continues along a bony strut that projects into the medullary cavity (Figure 7D).

### 3.2.3. Specimen FU2755, Callabonna CB2018-75, Indiv 1, F

In this specimen, the core does not penetrate the bone wall completely and, as a result, only the outer part of the cortex is preserved. In this region, an OCL with at least ~6–7 LAGs is preserved. Below this band of tissue, the cortex consists of laminar FBL bone tissue with circumferentially organised vascular canals.

### 3.2.4. Specimen SAM P.53833, Callabonna Geny10, ?F

This specimen preserves a fairly thick layer of cortical bone tissue. A distinct OCL is evident along the peripheral margin of the bone wall, and at least two LAGs are evident therein (Figure 7E). Below the OCL, the bone is mainly primary tissue consisting of laminar-plexiform organised FBL bone tissue (Figure 7E,F). In the outer part of the compacta, the tissue appears to be more laminarly textured with a predominance of circumferentially oriented canals (Figure 7E). Several scattered secondary osteons are present throughout the compacta, but in localised parts of the cortex, a tract of secondary osteons extends from the peripheral margin to the endosteal region (Figure 7F); this is likely related to the linea intermuscularis caudalis. In places, an ICL is present, but overall, the endosteal margin is highly resorptive (Figure 7G). The entry of the nutrient foramen into the bone cortex is evident in Section 10FB, and there are distinctive changes in the orientation of the bone tissue around the foramen (Figure 7H).

### 3.2.5. Summary of the Femoral Histology

The four femora studied showed features related to ontogenetic status. Specimen FU2758 appears to be from a young individual, which has just begun to deposit lamellar bone tissue, but an OCL is not yet present in the compacta. In this bone, there are sparse secondary osteons visible. Specimen FU2756 appears to be a slightly older individual—a well-developed OCL is present, and a LAG occurs therein. Specimen SAM P.53833 has the most mature compacta, with an OCL with multiple LAGs, and compacta with evidence of much more secondary reconstruction. It should be noted, however, that this section intersects the linea intermuscularis caudalis and the nutrient foramen, and therefore, it is expected to show localised changes as a consequence. Specimen FU2755 appears to be the most mature of the four femora—it has five or six LAGs in the OCL, but we cannot decipher any more of the nature of the bone tissue, because of core failure.

### *3.3. Tarsometatarsi*

### 3.3.1. Specimen FU2750, Callabonna CB2018-98

Overall, the bone tissue appears richly vascularised, although it is apparent that the deeper cortex is much more vascularised than the outer compacta (Figure 8). Towards the outer part of the bone wall, there are two distinct growth marks in the compacta (Figure 8). Following the outer LAG, there is a distinct OCL, and near the margin of the bone, there appears to be a LAG present (Figure 8).

### 3.3.2. Specimen SAM P.53832, Callabonna, Geny 9B, M

This section of the TMT shows a distinct OCL in the outermost part of the cortex (Figure 9A). Except for the OCL, the rest of the compacta appears to be intensely secondarily remodelled (Figure 9A–C). Although there is a lot of secondary reconstruction, dense Haversian bone proportions are not reached in the mid-cortical regions, as there is still primary bone between neighbouring secondary osteons. However, towards the medullary cavity, the secondary reconstruction is more extensively developed, and it appears to reach

dense Haversian levels, but there are also several large unfilled erosion cavities visible in this area (Figure 9C). In places, a narrow layer of lamellar bone (ICL) is visible (Figure 9C).

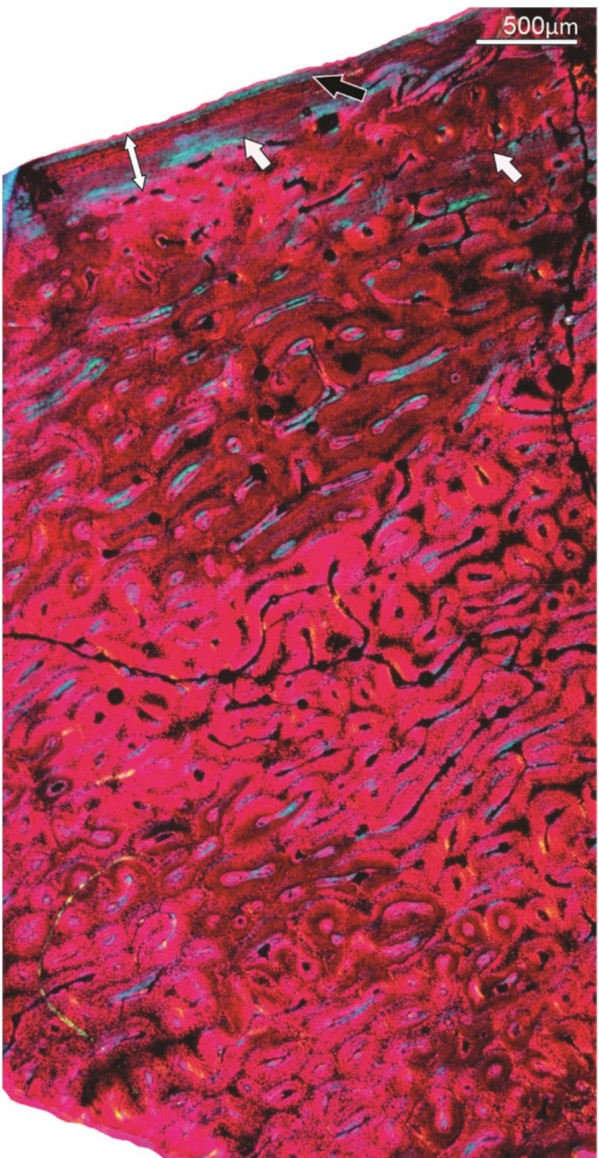

**Figure 8.** Specimen FU2750; Tarsometatarsus. Overview of the well-vascularised, primary compacta of the bone. In the peripheral region, a narrow OCL (double-headed arrow) and 2 narrow annuli (arrows) that precede it are visible. Subperiosteally, a LAG is present (black arrow).

### 3.3.3. Specimen SAM P.53831, Callabonna, Geny 9A, M

Overall, the bone tissue is as described for Specimen SAM P.53832, i.e., the compacta is extensively remodelled right up to the OCL, and the perimedullary region consists of dense Haversian bone.

### 3.3.4. Summary of Tarsometatarsi Histology

Similar to the femora and the tibiotarsi, the tarsometatarsi studied here show ontogenetic changes in the nature of the bone tissue. In this sample, the tarsometatarsus from the youngest individual (FU2750) has an OCL, but its compacta are still predominantly primary in nature, whereas in the tarsometatarsi from more mature individuals, except for the OCL, the compacta are intensively reconstructed and reaches dense Haversian characteristics in the deep cortex.

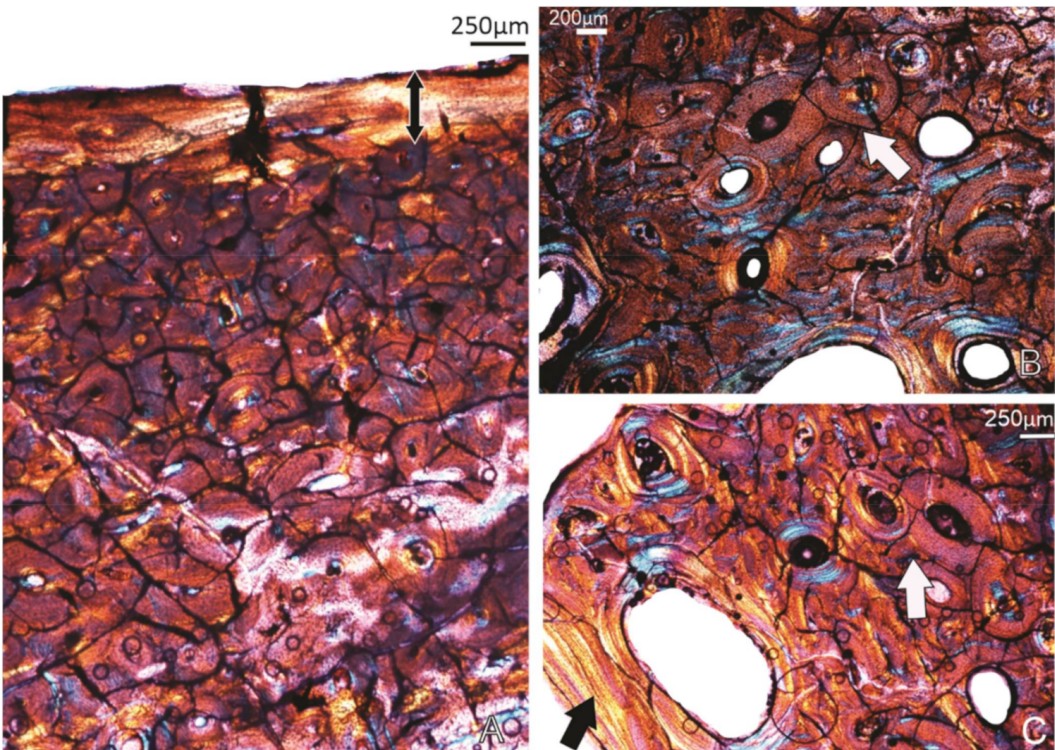

**Figure 9.** (**A–C**) Specimen SAM P.53832 TMT. (**A**) Except for the OCL (double-headed arrow), the compacta have been extensively reconstructed. (**B**) The dense development of secondary osteons (white arrow). (**C**) The perimedullary margin lined by an ICL (black arrow).

## 4. Discussion

### 4.1. Growth Pattern

In the *Genyornis* bones sampled, it is evident that during the earliest stages of growth, FLB was deposited. This tissue is typically formed in young fast-growing birds, e.g., *Struthio camelus* [16], *Sagittarius serpentarius* [16], *Aptenodytes patagonicus* [25] and *Calonectris leucomelas* [26]. This early bone has a number of longitudinally and reticular arranged primary osteons, and a large number of globular osteocytes in the woven bone matrix (see Figure 2C) [1]. In some of the bones, none of this early bone remains, whereas in a few bones where remodelling changes have not been extensive, the bone formed during early stages of ontogeny is preserved (e.g., tibiotarsus SAM P.54333), and we therefore have a continuous record of all the bone tissues formed during development. Often, this early bone is overlain by a laminar–plexiform bone, which tends to dominate the cortices of the tibiotarsus and the femora. In the tarsometatarsus, the predominant bone tissue is Haversian bone. Localised differences in the bone tissue were observed in response to the anatomy of the bones—for example, in the femur, in the region of muscle insertions, there is a radial tract of secondary osteons present (Figure 7F), and in the area where the nutrient foramen penetrates the bone, the bone tissue is organised so as to accommodate the foramen (Figure 7H).

In contrast to the bone rapidly forming during early ontogeny, in late stages of ontogeny, a distinct layer of lamellar bone tissue forms subperiosteally. This band of tissue, the OCL [1,18], marks the change to a slower rate of bone formation. Among many modern vertebrates, such a change is linked to the attainment of sexual maturity and the subsequent slow-down in growth, e.g., [1,27]. Thus, the occurrence of the OCL directly suggests that a slow-down in growth has been reached, which means that, thereafter, only slow accretionary growth will occur from this stage onwards. This seems to be the case in the extant kiwi [19], but in ducks, sexually immature ducks have been reported to show an OCL [27], and Watanabe [26] found the same in three species of water birds (*Calonectris leucomelas*,

*Ardea cinerea* and *Phalacrocorax capillatus*). Whether the OCL in *Genyornis* is linked to the attainment of sexual maturity is uncertain, but in the current study, it is apparent that *Genyornis* bones that morphologically appear to be from juveniles do not have an OCL (see later).

In several of the bones, narrow bands of lamellar bone tissue (annuli) were observed to have been deposited prior to the deposition of the OCL. The deposition of these periodic annuli interrupts the rapid phase of growth and reflect a slow-down in the overall rate of bone deposition [1,24], and they are generally thought to be formed annually in vertebrates, e.g., [17,27–29]. As most of our samples were cores, we cannot be certain that these annuli extend around the complete bone wall. In one specimen (tibiotarsus SAM P.54334), at least four such narrow annuli are observed, suggesting that this individual took at least 4 years before an OCL formed. Interestingly, in specimen SAM P.54334, the width of the annuli appears to be quite variable—even around the section (see Figure 4A,B). In parts of the compacta, some annuli appear as relatively wide bands of lamellar tissue. In most of the other individuals, there appears to only be about 1–2 growth marks evident in the compacta. Thus, it is evident that, unlike the largest of modern birds, ostriches (*Struthio camelus*), which weigh in at about 150 kg, *Genyornis* (estimated to have weighed about 250 kg) took more than a single year to reach skeletal maturity. *Vorombe titan*, the largest aepyornithid (and possibly the largest of all birds) [30] also took several years to reach skeletal maturity [5], and the extinct Dinornithiformes, such as the emeids, *Euryapteryx* and *Anomalopteryx*, as well as *Megalapteryx*, all experienced extended periods of cyclical growth to somatic maturity [20]. The extinct Mesozoic bird, *Gargantuavis*, was also found to have taken at least a decade to reach skeletal maturity [31]. Thus, it is evident that several large terrestrial birds experienced protracted growth to adult body size. It appears that other island birds which are not as large, such as the kiwi, *Apteryx* species [19], the dodo, *Raphus cucullatus* [28] and the solitaire, *Pezophaps solitaria* [32], also adopted slow, extended growth rates in response to reduced predation on the islands.

We were unable to identify any sex differences between the *Genyornis* femora studied. In *Dromornis stirtoni*, medullary bone [1], a tissue formed in female birds during ovulation was identified in several tibiotarsi [13], which verified their assignment as females. However, in the case of our *Genyornis* sample, medullary bone was not observed in any of the bones examined. It is likely that the birds were mired during a protracted drought, which may explain the lack of evidence of breeding (both in the form of hatchlings, and females with medullary bone), but it is also possible that our small sample size precluded the observation of sex-specific tissues.

### 4.2. Histological Variations Evident in Different Skeletal Elements

#### 4.2.1. Histological Differences among Bones without OCL

The tibiotarsus, specimen FU2759, and the femur, FU2758, were recovered from different individuals and, as such, we cannot directly compare their growth dynamics. However, on the basis of the histology evident, they appear to be from young individuals as they both do not have an OCL present. Furthermore, of these two bones, the tibiotarsus appears to be younger than the femur as rapidly formed FLB occurs subperiosteally in the tibiotarsus, whereas in the femur, a narrow band of lamellar bone tissue is present, indicating that the rate of bone deposition had begun to slow down. Interestingly, the surface texture of specimen FU2758 is clearly porous, and the crista trochanteris is also not fully formed, which further indicate that this is a young individual. The overall small diameter of the tibiotarsus specimen FU2759 also agrees with its young ontogenetic age assessment.

#### 4.2.2. Histological Variations Evident among Bones with OCL

Except for the bones mentioned above, all the other bones have a distinctive OCL, which means that the appositional growth had passed its most rapid phase of growth [16]. In some of the specimens, it is apparent that within the OCL, there are LAGs—assuming

that each LAG is formed annually, this indicates that some individuals are older than others, e.g., [31] (Table 2). This is further supported by an external morphology typical of fully adult birds; for example, as defined for dinornithiforms by Turvey and Holdaway [33], where femora all have the adult form of the condyles, all tibiotarsi have a fully ossified pons supratendineus and fully developed condyles with no sign of the synostosis between the tibia and proximal tarsal, and tarsometatarsi have no distinction of the fused metatarsi or distal tarsal.

**Table 2.** Summary of histology data for specimens studied. Spec. no., specimen number; OCL, outer circumferential layer; Lags, lines of arrested growth; ICL, inner circumferential layer; hav, Haversian; Resorpt. Perimedull., resorption cavities in the perimedullary region; Ontog., ontogenetic. Y denotes presence, N denotes absence and ? denotes uncertain.

| Specimen | Element | OCL | Lags in OCL | Annuli | ICL | Dense Hav Bone | Early Bone | Resorpt. Perimedull. | Ontog. State |
|---|---|---|---|---|---|---|---|---|---|
| SAM P.54334 | Tibiotarsus | Y | 4–5 | 3–4 | Y | Y | Y | Y | adult |
| SAM P.54333 | Tibiotarsus | Y | N | 1? | Y | N | Y | Y | young adult |
| SAM P.53833 | Femur | Y | 2 | Y | Y | Y | N | Y | adult |
| SAM P.53833 | Tibiotarsus | Y | 3 | 2? | N | Y | N | Y | adult |
| SAM P.53832 | Tarsometatarsus | Y | 2? | N | Y | Y | N | Y | adult |
| SAM P.53831 | Tarsometatarsus | Y | 2? | N | ? | Y | N | Y | adult |
| SAM P.53826 | Tibiotarsus | Y | 3–4 | 2? | ? | ? | ? | ? | adult |
| SAM P.25017 | Tibiotarsus | Y | 5 | 3 | Y | N | Y | Y | adult |
| FU2760 | Femur | Y | 1 | ? | Y | N | N | Y | adult |
| FU2759 | Tibiotarsus | N | N | N | N | N | Y | ? | immature |
| FU2758 | Femur | Y | N | N | Y | N | Y | Y | young adult |
| FU2756 | Tibiotarsus | Y | ~10 | 1 | N | N | Y | Y | mature |
| FU2750 | Tarsometatarsus | Y | 1 | 2 | ? | N | Y | Y | young adult |
| FU2755 | Tibiotarsus | ? | ? | ? | ? | N | Y? | ? | ? |
| FU2755 | Femur | Y | 6–7 | ? | ? | N | Y? | ? | mature |

Our sample of *Genyornis* bones provides evidence for different growth dynamics between individuals; although most individuals showed a periodic slow-down in growth in the form of narrow annuli, some individuals (e.g., tibiotarsus, SAM P.53833 from the Callabonna locality) had a wide band of lamellar tissue, suggesting that it experienced a particularly stressful period that was lengthy in duration. One specimen, SAM P.54334 (from Cooper Creek), showed four narrow annuli in the tibiotarsus, indicating that it had at least four periods of slowed growth, and two of these were longer in duration. The facts that, in some specimens, we find no annuli interrupting growth, and up to four in one individual, as well as the widely varying thickness of the annuli, suggest that *Genyornis* experienced variable growth dynamics, which may have been correlated with the particular environment during which the individuals were growing up. Such plasticity in growth appears to be a plesiomorphic trait inherited from their dinosaurian ancestors [1,34].

One of the main reasons for this discrepancy could be the fact that the specimens studied come from different localities, which, although they are not greatly separated, i.e., Billeroo Creek is perhaps 100 km from Lake Callabonna, which is 500 km at most from

Cooper Creek (Figure 1), they may have had different local ecologies. It is also possible that the specimens derive from slightly different time periods of the late Pleistocene, which makes it likely that they experienced different environmental conditions during their lives, i.e., they do not reflect a single contemporaneous population. The strikingly wide annulus present in the tibiotarsus specimen SAM P.53833 indicates that this individual experienced a prolonged stressed period when osteogenesis slowed down (Figure 6) [1,6]. However, once the conditions changed to a more favourable situation, osteogenesis recovered to a rapid rate, resulting in FLB tissue being formed. This is directly contrasted with the tarsometatarsus specimen FU2750, which had no annuli prior to the OCL formation. In the tibiotarsus specimen SAM P.54334 from Cooper Creek, two distinct wider-than-usual annuli are also observed (Figure 4A).

The Cooper Creek specimens (SAM P.25017, 54,333 and 54334) and Billeroo Creek specimens (FU2758, 2759) were deposited in fluvial sediments in a riverine situation, which does mean they had abundant water at their death. The Callabonna specimens were all trapped in the dried-out bed of a lake during a protracted drought.

### 4.2.3. Histological Differences among Specimens Recovered from the Same Site

Specimen SAM P.53832 and SAM P.53831 were both recovered from Lake Callabonna within 1 m of the other and facing the same direction, which suggests that they may have been trapped together. Interestingly, both these TMT show that they are mature adult individuals with a well-developed OCL and heavily reconstructed compacta. The overall dimension of these bones suggests that SAM P.53832 was the larger bird and likely to be a male, while SAM P.53831 was a probable female and was similar in size to SAM P.53833.

The tibiotarsi specimens, FU2756 and FU2755, were sampled from individuals that were recovered from Callabonna CB2018-75. The tibiotarsus from FU2756 has about 10 closely associated LAGs in the OCL, although we cannot be sure if any of them are part of double or triple LAGs which are known to occur in some vertebrates when conditions are recurrently unfavourable [1,27]. It must be noted that these lines do not interrupt the rapid phase of growth but are located in the outermost part of the compacta and are bone deposits that are responsible for the thickening or robustness of the bones (i.e., they do not contribute to the lengthening of the bone) [6]. As this tibiotarsus occurs in the boundary size range of male/female, these histology findings suggest that it is a mature female (rather than a young male). Unfortunately, the outer compacta of the tibiotarsus of FU2755 is not as well preserved, and we cannot determine whether or not it had passed its most rapid phase of growth. Its relatively small size suggests it is also a female individual.

### 4.3. Secondary Reconstruction

Secondary reconstruction was observed in all the skeletal elements, but compared to the femur and the tibiotarsi studied, the tarsometatarsus was the most extensively reconstructed element, with dense Haversian bone tissue present. This finding agrees with [35] that there is a proximodistal gradient in terms of secondary reconstruction with more distal elements being more extensively remodelled. Note that in the aepyornithids, the fibula was the most reconstructed element [5], but in the current study, fibulae were not sampled. It is possible that the high incidence of Haversian bone in the tarsometatarsus suggests that this element bears more weight and is subjected to more biomechanical forces than the tibiotarsi [36].

It is also apparent that the extent of secondary reconstruction in the compacta is age-related—the tarsometatarsus specimen FU2750 has an OCL but shows hardly any secondary reconstruction, whereas the other two tarsometatarsus studied have compacta that are completely remodelled right up to the OCL. These findings suggest that secondary reconstruction increases with age.

As in the aepyornithids [5], the tibiotarsi in *Genyornis* appears to provide the best record of growth and preserves most of the primary compacta. The tarsometatarsus is

useful in young adults, but older individuals show extensive secondary remodelling that removes the primary bone tissues, and hence the growth record.

## 5. Conclusions

*Genyornis* took more than a single year to reach sexual maturity, whereupon an OCL develops, indicating a change in the rate of osteogenesis.

Of the three skeletal elements studied, the tibiotarsus preserves the best record of growth for *Genyornis*.

The occurrence of several LAGs in the OCL indicates that it continued to accrete bone for several years to reach skeletal maturity. This further indicates that, in *Genyornis*, sexual and skeletal maturity were asynchronous, with sexual maturity preceding skeletal maturity.

*Genyornis* retained a plesiomorphic flexible growth strategy [1,15] and responded to prevailing environmental conditions at the time.

**Author Contributions:** A.C. and T.H.W. conceived the project and sampled the bones together; A.C. carried out the histological descriptions, took the photomicrographs and wrote the first draft of the MS. All authors have read and agreed to the published version of the manuscript.

**Funding:** This research was funded by Australian Research Grant, ARC Discovery Project DP180101913 "Extricating extinction histories at Lake Callabonna's megafauna necropolis", to T. Worthy, L. Arnold, and A. Chinsamy-Turan. AC is also supported by the National Research Foundation (NRF), grant number 117716. The APC was funded by the journal.

**Informed Consent Statement:** Not applicable.

**Data Availability Statement:** All data is within the manuscript.

**Acknowledgments:** Joshua van der Blerk is acknowledged for technical assistance with some of the thin section preparation. We thank Warren Handley for assistance with sampling the specimens. Three anonymous reviewers are thanked for their constructive comments.

**Conflicts of Interest:** The authors declare no conflict of interest.

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
