# Peer review of "Histovariability and Palaeobiological Implications of the Bone Histology of the Dromornithid, Genyornis newtoni"

_diversity, doi:10.3390/d13050219_

Round 1

Reviewer 1 Report

The authors studied the bone histology of the Pleistocene bird Genyornis newtoni to reconstruct different aspects of its biology. It is an interesting contribution, in which research is conducted properly and histological descriptions are easy to follow. Therefore, I recommend its publication once the authors address the minor comments that I detail below.

General comment

In the material and methods section of the manuscript, authors stated that “Genyornis newtoni shows a non-overlapping bimodal size distribution” in which females are smaller than males. However, they did not report any histological difference related to sex. They could have only found these differences in the femur, since it is the only bone in which both males and females were studied. How did you explain the lack of sex-related differences in the microstructure of this bone? Is it probably related to the sample size? Could it be that femur size does not vary that much between males and females? Would you expect to find sex-related differences in the histology of the tibiotarsus, since it is the bone that you used to infer the sex of the bones? I think it would be worth it to briefly address this issue in the manuscript, since sexing the different bones was part of the research.

Specific comments

Line 81 – 82 “Fifteen specimens of Genyornis were studied: eight tibiotarsi (TT), three tarsometatarsi (TMT), and four femora”: please provide an abbreviation for the femora, as you did for the other limb bones.

Line 95 – 96 “They were then sectioned along the midline, and divided into two blocks, A and B”: please explain in more detail why each sample was divided into two blocks. Were the samples too big to be prepared using only one slide? Did these blocks represent different anatomical regions (e.g. block A: anterior, block B: posterior)?

Line 111 “These revealed two individuals that were much were larger than those…”: Please remove the second “were” of this sentence.

Lines 130 -137: Please review the reference to the different panels of figure 2. I think authors wanted to refer to “Figure 2A, 2B” in line 132, and “Figure 2C” in lines 135-136.

Line 167: There is a bracket missing before Fig. 4A. Also, the authors probably wanted to refer to Fig. 4B and not Fig. 4A. Please check.

Line 206 “so the most recently ormed bone tissue cannot be described”: Please change the word ormed by formed.

Line 267 “Overall, the bone wall not very well preserved”: I think there is a verb missing between “wall” and “not”.

Line 275 “The compacta comprises of predominantly of short circumferentially”: Please remove one “of”.

Line 349 -354 Summary of tarso metatarsi histology: Please indicate in brackets the specimen number of what you consider “younger” and “older” individuals, as you did in the summary of the other limb bones.

Line 358 – 359 “This tissue is typically formed in young fast growing birds e.g. [14,25,26]”: Please indicate the name of these other birds presenting FLB.

Line 379 – 381 “Thus, the occurrence of the OCL in Genyornis directly suggests that a slow down in growth has been reached which means that thereafter only slow accretionary growth will occur from this stage onwards”. I suggest that you move this sentence after the one ending as “and the subsequent slow down in growth e.g. [1,27]” in line 376. This will facilitate the reading and understanding of the paragraph.

Line 389: the authors forgot to provide an example after “(e.g. )”. Please provide one.

Lines 412-414 “Furthermore, of these two bones, it is apparent that the former is younger than the latter since rapidly formed FLB occurs subperiosteally in the tibiotarsus, whereas in the femur, a narrow band of lamellar bone tissue is present”: Authors may need to explain better this inference. At a specific ontogenetic moment (e.g. two years old), some bones could continue growing while others had already stopped. Hence, the identification of an incipient OCL in the femur but not in the tibiotarsus might not necessarily imply that the former bone belongs to an older individual. Maybe, it just means that the femur and the tibiotarsus present different growth patterns.

Lines 450 – 451 “It is also possible that they each sample slightly different time periods of the late Pleistocene”. I think there is a word missing in this sentence.

Figures

Figures 3, 4, 5, 6, 7 and 9 do not appear complete in the PDF. This could be a problem of the PDF-generation software, but please check that dimensions are correct.

Figure 3: I would suggest the authors to include some labels to better show the different FLB tissues (e.g. reticular, plexiform-circumferential) and histological structures (e.g. SO). They should also indicate in the caption that the black arrow points to the ICL.

Figure 5: I do not understand why this figure includes two different individuals, since they do not belong to the same palaeontological site and all previous tibiotarsus figures only illustrate one individual. I suggest that you prepare a new figure only including images from specimen SAM P.25017. Also, please indicate in the figure caption that the arrows in panel A and E indicate LAGs.

Figure 6: Authors forgot to draw the double headed arrow in panel A. Also in this panel, the black arrow pointing to the wide band of lamellar bone and the black arrow pointing to the resorption cavity are of similar size. Please modify one of them to fit the information in the figure caption.

Tables

Table 2: I recommend the authors to always use the same symbols to summarize the histology data (e.g. X = presence). What do the blank spaces mean? Are they equivalent to the question marks?

Author Response

The reviewer is thanked for the review.  Attached is a document detailing our response to each comment. 

Reviewer 2 Report

The manuscript performs an analysis of the histology of some elements of the bird Genyornis. Methodologically, the analysis is classic and is carried out properly. The description of characters and the discussion is adjusted to the information provided. The conclusion that is obtained is immediate from the results, but it is interesting since it corroborates that Genyornis presents a type of growth that is unusual in birds, but that it is expected in birds of its size and under those environmental conditions. Except for small formal problems, some of which are noted in the list that follows, I think the manuscript can be published in its current state

Some matters to review:

Line 31: Is it possible to rephrase to avoid the combination "deduct" "deductions"?

Line 191: The figure appears to be cut off (it may just be a problem with the file version)

Line 192: reference to medullary margin (m) is missing

Line 198: The figure appears to be cut off (it may just be a problem with the file version)

Line 222: The figure appears to be cut off (it may just be a problem with the file version)

Line 223: reference to medullary margin (m) is missing

Line 229-230: Control font type

Line 257: The figure appears to be cut off (it may just be a problem with the file version)

Line 293: If there are no more references to the linea intermuscularis caudalis, the acronym (lic) would not be necessary

Line 309: The figure appears to be cut off (it may just be a problem with the file version)

Line 341: The figure appears to be cut off (it may just be a problem with the file version)

Line 369: Confirm that it refers to figure 5F

Line 371: Does figure 5H exist? (the image could be in the part that does not appear in my copy) Are you referring to figure 7H?

Line 372: Separate the words: “forming during”

Line 505-506: Separate words

From line 525 to end: Control the font type and size

Line 575: Reference must be Chinsamy and Raat

Author Response

The reviewer is thanked for the comments made.  Attached is a document that details our response to the comments. 

Reviewer 3 Report

A nice little paper with significant results - several times in the discussion there is an assumption that the large individuals may be male but it is not stated explicitly why this is assumed and I assume that this paper is the only test of this hypothesis?? - I think this needs to be explicitly referenced in the introduction (even if it just saying that all Galloanseriiformes have male larger SD) and this could be stated as a hypothesis which this paper test with the results being.... 

My only comments are a few very minor language issues:

182 "Unfortunately, a full core did not result from this specimen" should be "we were unable to retrieve a full core from this specimen..."

211 The well preserved compacta of this tibiotarsus shows a well vascularised compacta with a distinctly wide OCL" should be "The well-preserved compacta of this tibiotarsus shows that it is heavily vascularised..."

307 "has between 5-6 LAGs" should be "has between 5 and six LAGs" or just "has 5-6 LAGS"

Author Response

The reviewer is thanked for the comments made. Attached is  detailed response to the comments.
